# Effect of Dietary Compound Acidifiers Supplementation on Growth Performance, Serum Biochemical Parameters, and Body Composition of Juvenile American Eel (*Anguilla rostrata*)

**Mingliang Zhang †, Xinyi Wu † and Shaowei Zhai \***

Engineering Research Center of the Modern Industry Technology for Eel, Ministry of Education,
Fisheries College of Jimei University, Xiamen 361021, China

\* Correspondence: zhaisw@jmu.edu.cn

† These authors contributed equally to this work.

**Abstract:** As growth-promoting feed additives, compound acidifiers (CAs) have been widely reported in many farmed fish species, whereas there is little information on the application of CAs in the eel diet. The present trial was conducted to evaluate the supplementation effects of CAs on growth performance, serum biochemical parameters, and body composition of the juvenile American eel (*Anguilla rostrata*). The CAs supplementation levels in the diet of American eel were 0, 2.0, 3.0, 4.0, and 5.0 g/kg, respectively. The trial lasted for 12 weeks. The most beneficial effects of CAs on promoting growth performance, decreasing levels of serum lipid, modifying parameters of the liver function and immune function in the serum, and increasing the utilization of calcium and phosphorus in the diet were observed in the fish fed with 4.0 g/kg CAs. Dietary CAs supplementation had no significant effects on the body composition of whole fish, whereas 5.0 g/kg of dietary CAs supplementation exerted negative effects on growth performance, serum biochemical parameters, and utilization of calcium and phosphorus. The recommended level of CAs in the diet was 3.5–3.7 g/kg to achieve the optimal weight gain and feed conversion ratio for the juvenile American eel.

**Keywords:** compound acidifiers; growth performance; serum lipid; immune function; body composition; *Anguilla rostrate*; calcium accumulation; phosphorus accumulation

## 1. Introduction

Acidifiers are now widely used in the livestock and poultry feed industry as an alternative to antibiotic growth promoters. The acidifiers supplemented in animal feeds are mainly compound acidifiers (CAs), which are generally composed of certain proportions of organic acids or the salts of these organic acids [1,2]. Because they combine the different antimicrobial activities of certain organic acids, CAs usually have broader antibacterial activity than a single acidifier [3,4]. In addition, the different components of CAs may also synergize well with each other to maximize their effects and benefits [1,2,5]. Therefore, CAs have better effects than a single acidifier, and many studies of CAs have been reported in livestock and poultry feeds. The results of these earlier studies highlight CAs' role in growth promotion, immune enhancement, improved digestion, nutrient digestibility, and intestinal health [1,6].

Recently, aquafeeds have also started to be fortified with CAs to improve the growth and health of fish [6]. Studies have shown the growth-promoting effects of appropriate supplementation levels of CAs in the diets of Nile tilapia (*Oreochromis niloticus*) [7,8], striped mullet (*Mugil cephalus*) [5], African catfish (*Clarias gariepinus*) [5], Mrigal carp (*Cirrhinus mrigala*) [9], and Japanese sea bass (*Lateolabrax japonicus*) [4]. In addition, the CAs can improve intestinal health and disease resistance in olive flounder (*Paralichthys olivaceus*) [10], and ameliorate the growth and immunity of yellowfin bream (*Acanthopagrus latus*) fed

high plant protein diets [11]. However, dietary CAs supplementation did not significantly improve the growth of rainbow trout (*Oncorhynchus mykiss*) [12] or yellow catfish (*Pelteobagrus fulvidraco*) [13]. These results suggest that the beneficial effects of dietary CAs supplementation may vary considerably in different fish species. Therefore, the specific role of CAs in aquaculture is not yet fully established, and their properties merit further exploration in other fish species [4].

Eel, one of the most common freshwater cultured fish in the world, is rich in vitamins, proteins, and unsaturated fatty acids, and has important medicinal value; thus, it is known as "ginseng in water" [14]. After more than 40 years of development, the eel industry in China has formed an export-oriented industrial chain, including eel culture, feed production, the processing of eel products, and export integration, which has made important contributions to the development of the Chinese fisheries economy [14]. With the natural stocks of European eel (*Anguilla anguilla*) and Japanese eel (*Anguilla japonica*) declining sharply, American eel (*Anguilla rostrata*) has become one of the main eel farming species in southeastern China [2]. Although numerous studies have investigated the beneficial roles of CAs on growth and health in many cultured fish species, little is known about the effects of CAs on the American eel. Therefore, the present study aimed to evaluate the effects of CAs on growth performance, serum biochemical parameters, and body composition of the juvenile American eel.

## 2. Materials and Methods

### 2.1. Trial Design and Diets

Six hundred juvenile American eels with similar body weight ($11.00 \pm 0.03$ g per fish) were randomly divided into five treatments and fed diets with CAs levels of 2.0, 3.0, 4.0, and 5.0 g/kg, respectively. The five treatment groups were the CAs0, CAs2, CAs3, CAs4, and CAs5 groups, respectively. There were four replicates in each treatment group with 30 fish per replicate. The experimental period was 12 weeks. The CAs in this trial were manufactured by Nutreco B.V., the Netherlands, and the guaranteed values of main components in CAs were formic acid $\geq$ 12%, ammonium formate $\geq$ 12%, acetic acid $\geq$ 4.4%, propionic acid $\geq$ 5.2%, sorbic acid $\geq$ 0.4%, lactic acid $\geq$ 0.5%, citric acid $\geq$ 0.1%, and caproic acid + caprylic acid + capric acid + lauric acid $\geq$ 8.6%.

The basic diet was a commercial powdered diet for black eels (Fuzhou Xinruiyi Industrial Co., Ltd., Fuzhou, China), and mainly contained white fish meal, brown fish meal, starch, extruded soybean, yeast powder, and compound premix. The nutrient composition of the commercial diet was crude protein 46.16%, crude lipid 7.31%, crude ash 14.62%, moisture 5.14%, calcium (Ca) 3.74%, and phosphorus (P) 2.17%. The commercial diet was served as the control diet with no CAs supplementation, and trial diets were prepared with the inclusion of CAs of graded levels.

### 2.2. Trial Fish Management

A total of 1200 juvenile American eels were obtained from Fujian Jinjiangzhiman Aquatic Technology Co., Ltd., Zhangzhou, China. Before the trial, all fish were acclimatized in two 1200 L PVC tanks (filled with approximately 800 L of water). The surface of PVC tanks was covered with black shade netting to prevent the eels from being exposed to light stress. During the acclimatization period, all fish were fed the commercial diet twice daily at 06:30 and 18:30. The powder diet was mixed with a 1:1.2 volume of water to form a dough shape, and then the dough feed was placed on a feeding table for fish. The uneaten feed was siphoned out 20–25 min after feeding. After 4 weeks of acclimation, six hundred juvenile American eels with similar body weight were selected and cultured in separate tanks (approximately 400 L of water injection) with a water recirculation device for formal trial. The following water quality parameters were maintained during the formal trial period: temperature 25.1–27.2 °C, pH 7.0–7.8, dissolved oxygen 7.1–9.0 mg/L, total ammonia nitrogen < 0.65 mg/L, and nitrite < 0.055 mg/L.

### 2.3. Sample Collection and Analysis

At the end of the feeding trial, all fish in each tank were deprived for 24 h, and then fish were anesthetized with eugenol (0.1 mg/L), weighed, and counted to calculate the growth performance parameters. The blood of ten eels from each tank was sampled, treated, and mixed as one sample before analysis of serum biochemical parameters according to the procedure of Zhai et al. [15]. Four fish were randomly selected from each tank and stored frozen ($-20\ °C$) to determine the body composition and Ca and P contents in whole fish.

For analysis of serum biochemical parameters, commercial kits produced by Nanjing Jiancheng Bioengineering Institute (Nanjing, China) were used to measure blood urea nitrogen (BUN), Ca, P, triglyceride (TG), total cholesterol (TC), high-density lipoprotein cholesterol (HDL-C), low-density lipoprotein cholesterol (LDL-C), glutamic pyruvic transaminase (GPT), glutamic-oxalacetic transaminase (GOT), acid phosphatase (ACP), alkaline phosphatase (AKP), lysozyme (LZM), complement3 (C3), and immunoglobulin M (IgM) in serum of the American eel.

For measurements of the proximate composition of whole fish and contents of calcium (Ca) and phosphorus (P) in the whole fish and trial diets were determined by the AOAC method [16]. The moisture content was determined by drying the samples to a constant weight at 105 °C in a drying oven. The ash content was determined by incineration at 550 °C. The crude protein content was obtained by measurement of nitrogen (N × 6.25) using the Kjeldahl method. Crude lipid content was determined by the Soxhlet method. The Ca content was determined by ethylenediaminetetraacetic acid titration and the P content was determined by the colorimetric molybdenum yellow method. The accumulation rate of dietary Ca or P was expressed as the accumulation weight of Ca or P in whole fish dividing by the intake weight of dietary Ca or P.

### 2.4. Statistical Analysis

The trial results data are presented as mean $\pm$ SD and subjected to one-way ANOVA using the SPSS 23.0 statistical software (SPSS, Chicago, IL, USA). Before analysis, data were subjected to the homogeneity of variance test. For the data expressed as percentages or ratios, square arcsine transformation was applied before further statistical analysis. Duncan's multiple range test was used to compare the difference in mean values among different treatment groups after overall differences were found to be significant ($p < 0.05$).

## 3. Results

### 3.1. Growth Performance

The effects of dietary CAs supplementation on the growth performance of juvenile American eel are shown in Table 1.

As shown in Table 1, the FFW, WGR, SGR, FR (except for the CAs2 group), FI, and PER of the CAs supplementation groups were significantly higher than those of the CAs0 group ($p < 0.05$), and those parameters in the CAs4 group were highest among all the groups. Compared with the CAs0 group, the FCR of CAs supplementation groups was significantly decreased ($p < 0.05$), and the FCR of the CAs4 group was the lowest among all the groups. There was no significant difference in the SR among all the groups ($p > 0.05$). As shown in Figure 1, there was a significant quadratic trend regarding the increasing CAs supplementation levels with WGR or FCR. The quadratic regression equation between the CAs levels and WGR was $Y = -3.5403X^2 + 26.405X + 105.79$ ($R^2 = 0.7438$). The optimum level of dietary CAs supplementation was 3.7 g/kg (Figure 1A) to achieve the maximum WGR. The quadratic regression equation between the CAs levels and FCR was $Y = 0.0196X^2 - 0.1376X + 1.702$ ($R^2 = 0.8222$). The optimum level of dietary CAs supplementation was 3.5 g/kg (Figure 1B) to achieve the minimum FCR.

**Table 1.** Growth performance of juvenile American eel in different CAs supplementation groups.

| Items | CAs0 Group | CAs2 Group | CAs3 Group | CAs4 Group | CAs5 Group |
|---|---|---|---|---|---|
| IFW (g/fish) | 11.03 ± 0.04 [a] | 11.01 ± 0.03 [a] | 11.00 ± 0.01 [a] | 10.99 ± 0.02 [a] | 10.99 ± 0.03 [a] |
| FFW (g/fish) | 23.07 ± 0.50 [a] | 25.67 ± 0.43 [b] | 27.73 ± 0.86 [c] | 30.02 ± 1.19 [d] | 26.55 ± 0.24 [b] |
| WGR (%) | 109.26 ± 4.76 [a] | 133.25 ± 3.75 [b] | 152.05 ± 7.55 [c] | 173.20 ± 11.11 [d] | 139.70 ± 3.96 [b] |
| SGR (%/d) | 0.88 ± 0.03 [a] | 1.01 ± 0.02 [b] | 1.10 ± 0.04 [c] | 1.20 ± 0.05 [d] | 1.04 ± 0.02 [b] |
| FCR | 1.69 ± 0.07 [c] | 1.54 ± 0.02 [b] | 1.48 ± 0.02 [b] | 1.39 ± 0.07 [a] | 1.54 ± 0.03 [b] |
| FR (%) | 1.42 ± 0.02 [a] | 1.46 ± 0.01 [ab] | 1.52 ± 0.05 [c] | 1.54 ± 0.02 [c] | 1.50 ± 0.01 [bc] |
| FI (g/fish) | 20.32 ± 0.10 [a] | 22.51 ± 0.47 [b] | 24.72 ± 1.44 [c] | 26.42 ± 0.43 [d] | 23.75 ± 0.18 [c] |
| PER (%) | 128.44 ± 5.17 [a] | 141.13 ± 1.61 [b] | 146.69 ± 1.60 [b] | 155.98 ± 7.64 [c] | 141.18 ± 2.89 [b] |
| SR (%) | 100 ± 0.00 [a] | 100 ± 0.00 [a] | 100 ± 0.00 [a] | 100 ± 0.00 [a] | 99.17 ± 1.67 [a] |

The values are means ± SD, n = 4. [a,b,c,d] Values with different superscripts in the same row are significantly different ($p < 0.05$). IFW (initial fish weight, g/fish) = initial fish weight (g)/initial number of fish. FFW (final fish weight, g/fish) = final fish weight (g)/final number of fish. WGR (weight gain rate, %) = 100 × [final fish weight (g)–initial fish weight (g)]/initial wet weight (g). SGR (Specific growth rate, %/d) = [(ln final fish weight (g/tank) − ln initial fish weight (g/tank))/time period (day)] × 100. FCR (feed conversion ratio) = feed consumed (g)/weight gain (g). FR (Feeding rate, %) = 100 × feed consumed (g)/[(final fish weight (g) + initial fish weight (g))/2]/time period (day). FI (feed intake, g/fish) = feed consumed (g)/number of fish in each tank. PER (protein efficiency rate, %) = 100 × weight gain (g)/protein intake (g). SR (survival rate, %) = 100 × (final number of fish/initial number of fish).

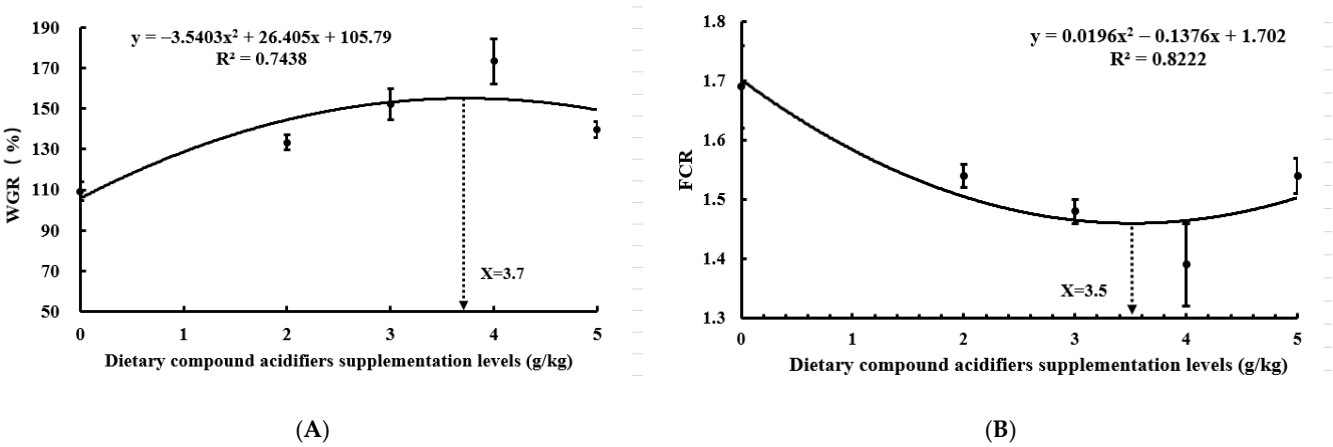

**Figure 1.** The quadratic regression between dietary CAs supplementation levels and WGR (**A**) or FCR (**B**).

### 3.2. Serum Biochemical Parameters

The effects of dietary CAs supplementation on serum biochemical parameters of juvenile American eel are shown in Table 2.

Compared with the CAs0 group, the BUN level of the CAs4 group was significantly lower than that of other groups ($p < 0.05$), and there was no significant difference between other groups except the CAs4 group ($p > 0.05$). Levels of Ca in the groups with CAs supplementation (except the CAs5 group) were significantly increased in comparison with the CAs0 group ($p < 0.05$). Levels of P in the groups with CAs supplementation (except the CAs2 group) were significantly increased in comparison with the CAs0 group ($p < 0.05$). There was no significant difference in the levels of Ca and P among the groups with CAs supplementation ($p > 0.05$). The levels of TG, TC, and LDL-C and activities of GOT and GPT in the CAs supplementation groups were significantly decreased ($p < 0.05$). The HDL-C levels in CAs2, CAs3, and CAs4 groups were significantly increased ($p < 0.05$), and there was no significant difference between the groups with CAs supplementation ($p > 0.05$). The AKP activity was significantly increased in CAs4 and CAs5 groups in comparison with those of CAs0, CAs2, and CAs3 groups ($p < 0.05$), and there was no significant difference in AKP activity between CAs4 and CAs5 groups ($p > 0.05$). The ACP activity only in the CAs4 group was significantly increased in comparison with that of the CAs0 group, CAs2, and CAs5 groups ($p < 0.05$). Compared with the CAs0 group, the levels of IgM, C3, and LZM

activity were significantly increased in the groups with CAs supplementation ($p < 0.05$), and their values in the CAs4 group were highest among all the groups.

**Table 2.** Serum biochemical parameters of juvenile American eel in different CAs supplementation groups.

| Items | CAs0 Group | CAs2 Group | CAs3 Group | CAs4 Group | CAs5 Group |
|---|---|---|---|---|---|
| BUN (mmol/L) | 10.19 ± 0.58 [b] | 9.93 ± 0.47 [b] | 9.33 ± 0.71 [ab] | 8.52 ± 0.92 [a] | 9.52 ± 0.34 [ab] |
| Ca (mmol/L) | 1.59 ± 0.03 [a] | 1.71 ± 0.09 [b] | 1.73 ± 0.11 [b] | 1.76 ± 0.06 [b] | 1.70 ± 0.07 [ab] |
| P (mmol/L) | 4.00 ± 0.29 [a] | 4.37 ± 0.48 [ab] | 4.57 ± 0.27 [b] | 4.71 ± 0.31 [b] | 4.68 ± 0.08 [b] |
| TG (mmol/L) | 7.50 ± 0.14 [d] | 4.86 ± 0.53 [b] | 4.29 ± 0.08 [a] | 4.16 ± 0.08 [a] | 6.36 ± 0.24 [c] |
| TC (mmol/L) | 30.87 ± 2.23 [b] | 26.09 ± 3.39 [a] | 22.58 ± 2.07 [a] | 22.40 ± 1.15 [a] | 25.15 ± 3.05 [a] |
| HDL-C (mmol/L) | 5.26 ± 0.35 [a] | 6.12 ± 0.66 [b] | 6.45 ± 0.60 [b] | 6.40 ± 0.37 [b] | 5.66 ± 0.48 [ab] |
| LDL-C (mmol/L) | 8.39 ± 0.97 [b] | 6.99 ± 0.87 [a] | 6.04 ± 0.83 [a] | 5.69 ± 0.91 [a] | 6.30 ± 0.74 [a] |
| GPT (U/L) | 7.34 ± 0.24 [e] | 5.22 ± 0.45 [d] | 3.72 ± 0.27 [b] | 1.37 ± 0.54 [a] | 4.55 ± 0.42 [c] |
| GOT (U/L) | 14.91 ± 0.58 [d] | 8.08 ± 0.48 [b] | 6.37 ± 0.33 [a] | 6.13 ± 0.83 [a] | 10.95 ± 1.48 [c] |
| AKP (U/L) | 3.15 ± 0.47 [a] | 3.29 ± 0.34 [a] | 3.32 ± 0.41 [a] | 4.54 ± 0.40 [b] | 4.04 ± 0.39 [b] |
| ACP (U/L) | 7.68 ± 0.59 [ab] | 7.72 ± 0.19 [ab] | 8.81 ± 0.48 [bc] | 9.02 ± 0.94 [c] | 7.24 ± 1.29 [a] |
| LZM (U/mL) | 167.59 ± 5.14 [a] | 204.24 ± 10.88 [b] | 258.96 ± 7.57 [c] | 276.06 ± 12.09 [d] | 215.96 ± 10.16 [b] |
| C3 (µg/mL) | 2417.33 ± 102.53 [a] | 2561.25 ± 54.23 [b] | 2565.33 ± 13.02 [b] | 2732.04 ± 17.91 [c] | 2599.58 ± 106.76 [b] |
| IgM (µg/mL) | 868.75 ± 67.99 [a] | 1016.97 ± 39.73 [b] | 1236.36 ± 81.50 [c] | 1381.47 ± 62.85 [d] | 1214.36 ± 134.93 [c] |

The values are means ± SD, n = 4. [a,b,c,d] Values with different superscripts in the same row are significantly different ($p < 0.05$). BUN = blood urea nitrogen, Ca = calcium, P = phosphorus, TG = triglyceride, TC Total cholesterol, HDL-C = high-density lipoprotein cholesterol, LDL-C = low-density lipoprotein cholesterol, GPT = glutamic pyruvic transaminase, GOT = glutamic-oxalacetic transaminase, ACP = acid phosphatase, AKP = alkaline phosphatase, LZM = lysozyme, C3 = complement3, IgM = immunoglobulin M.

### 3.3. Body Composition of the Whole Fish

The effects of dietary CAs supplementation on the body composition of juvenile American eel are presented in Figure 2. No significant differences in the contents of moisture, crude protein, crude lipid, and ash were observed among all groups ($p > 0.05$).

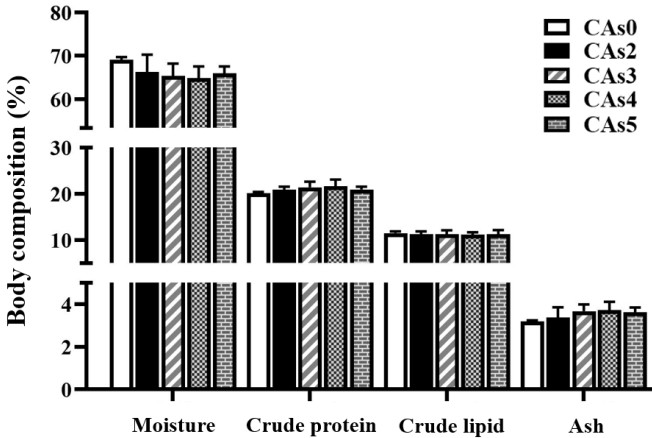

**Figure 2.** Body composition of juvenile American eel in different CAs supplementation groups. No superscript on the bar chart means no significant difference ($p > 0.05$).

### 3.4. Macroelement Content in the Whole Fish

The content of macroelements (Ca and P) in the whole fish is shown in Figure 3. Compared with the CAs0 group, the Ca content in the groups with CAs supplementation (except the CAs2 group) ($p < 0.05$), and P content only in the CAs3 and CAs4 groups was significantly increased ($p < 0.05$). No significant difference in P content was observed in the CAs2 and CAs5 groups in comparison with the CAs0 group ($p > 0.05$). Furthermore, the contents of Ca and P showed a general increasing trend from the CAs0 group to the CAs4

group and a decreasing trend from the CAs4 group to the CAs5 group, and the contents of Ca and P in the CAs4 group were highest among all the groups.

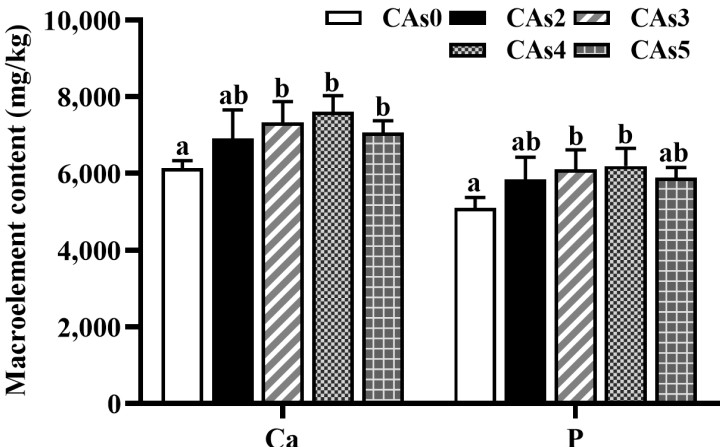

**Figure 3.** The contents of Ca and P in the whole fish of juvenile American eel in different CAs supplementation groups (wet sample basis). [a,b] Different superscripts on the bar mean the significant differences ($p < 0.05$).

### 3.5. Accumulation Rate of Dietary Ca and P

The accumulation rates of dietary Ca and P are shown in Figure 4. Compared with the CAs0 group, the accumulation rates of Ca and P were significantly increased in the groups with CAs supplementation ($p < 0.05$). The Ca accumulation rate in the CAs4 group was significantly higher than that of the CAs0, CAs2, and CAs5 groups ($p < 0.05$). There was no significant difference in P accumulation rate among the groups with CAs supplementation ($p > 0.05$). Furthermore, the accumulation rates of Ca and P showed a general increasing trend from the CAs0 group to the CAs4 group and a decreasing trend from the CAs4 group to the CAs5 group, and the accumulation rate of Ca or P in the CAs4 group was the highest among all the groups.

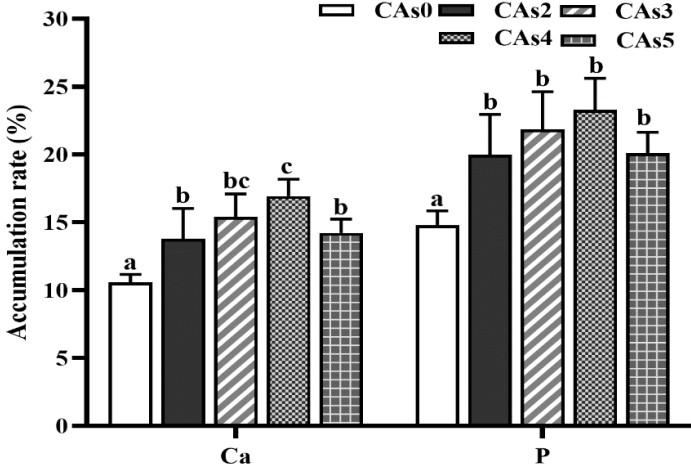

**Figure 4.** The accumulation rates of Ca and P of juvenile American eel in different CAs supplementation groups. [a,b,c] Different superscripts on the bar mean the significant differences ($p < 0.05$).

### 4. Discussion

In the present trial, the growth parameters of juvenile American eel, including FFW, WGR, SGR, FCR, FR, FI, and PER, were improved by dietary CAs supplementation, and the growth promotion effect of CAs was obvious. Similar to the results of the present trial, dietary supplementation with 5.0–15.0 g/kg of CAs (formic acid and calcium propionate) in *Cirrhinus mrigala* [9], 5.0–15.0 g/kg of CAs (malic acid and oxalic acid) in Nile tilapia [8],

1.0–2.0 g/kg CAs (formic acid, propionic acid, and calcium propionate) in Nile tilapia [7], and 2 or 4 g/kg CAs (citric acid, lactic acid, and phosphoric acid) in Japanese sea bass [4] were found to improve the growth and feed utilization of those fish species. Additionally, 10.0 g/kg of CAs (sodium propionate and sodium acetate) in high plant protein diets could also improve the growth of yellowfin seabream [11]. However, some studies observed that the parameters of growth performance were not improved by 5 g/kg of dietary CAs (fumaric acid, benzoic acid, and fumaric acid, benzoic acid, and 2-hydroxy-4-methylthio butanoate acid) in yellow catfish [13], 4 g/kg CAs (CAs A: formic acid, ammonium formate, and propionic acid or CAs B: benzoic acid, fumaric acid and hydroxy analog of methionine) in olive flounder [10], and 4 g/kg CAs (ammonium formate, formic acid, vegetable fatty acids, propionic acid, and acetic acid) in Nile tilapia [17].

Biochemical parameters of blood are generally used for assessing the health status of fish. Urea is the end product of purine metabolism, and its level in blood can reflect protein metabolism and the amino acid balance of the body, and lower levels of BUN indicated a possible improvement in protein utilization efficiency and increase in nitrogen deposition rate [18]. In the present study, dietary supplementation with 4.0 g/kg CAs lowered the BUN level significantly, which suggests that CAs can improve the dietary protein utilization for juvenile American eel. Similarly, Zhao et al. [19] reported that the dietary supplementation with 0.25–1.00 g/kg sodium butyrate in yellow catfish can significantly reduce levels of BUN before and after ammonia nitrogen stress. Dietary CAs supplementation increased the levels of Ca and P in juvenile American eel. This may contribute to the effects of CAs on promoting the dissolution of minerals such as insoluble tricalcium phosphate in feed and improving the utilization of dietary Ca and P in the intestine [3,20]. The above results indicate that adequate CAs supplementation may improve the nutrient absorption of the American eel.

The levels of TG, TC, HDL-C, and LDL-C in serum are important parameters of lipid metabolism catabolism and transport in the liver of the animals [21]. In this trial, certain dietary CAs supplementation significantly decreased levels of TG, TC, and LDL-C and increased the HDL-C levels in the serum of trial fish. These results suggest that dietary CAs may have hypolipidemic effects on juvenile American eel. Similar to the results of this experiment, dietary supplementation with 9 g/kg potassium diformate can significantly reduce the levels of both TG and TC and increase the HDL-C level in the serum of golden pompano [22]. This may be because the acetic acid in the CAs can induce the activation of the AMPK signaling pathway to affect lipid oxidative decomposition and de novo synthesis in the liver [23]. Additionally, the propionic acid in CAs may activate the PPARγ signaling pathway in the liver and adipose tissue to reduce the level of triglycerides in the liver and blood lipids levels [24,25].

GOT and GPT are mainly located in the liver and reflect the health and function of the liver by regulating the transferring function of α-amino acids to α-keto acids [8,26]. A large amount of GOT and GPT can be released into the blood, mostly during liver damage, so the activities of GOT and GPT can be used as important indicators to evaluate the degree of liver injury [8,26]. In this trial, dietary CAs supplementation significantly lowered the activities of GOT and GPT in serum, which suggests that dietary CAs supplementation might be beneficial for improving the liver health of juvenile American eel. This phenomenon was similar to the results of dietary supplementation with 2.0 g/kg sodium butyrate of Gold pomfret (*Trachinotus Ovatus*) [27] and 5.0–15.0 g/kg of CAs (CAs A: malic acid and oxalic acid or CAs B: calcium lactate and sodium acetate) in tilapia [8]. In addition, 5.0 and 10.0 g/kg potassium diformate could significantly decrease the activities of GOT and GPT in tilapia fed with fish meal free diets [26]. The improvement in liver function may be related to the ability of the CAs to modulate the immunity function and inhibit liver inflammation, and the translocation of intestinal bacteria to the liver [27,28]. However, 4.0 g/kg of dietary CAs supplementation (CAs A: formic acid, ammonium formate, and propionic acid or CAs B: benzoic acid, fumaric acid, and hydroxy analog of methionine) exerted no significant effects on activities of GOT and GPT in olive flounder [10].

ACP is a typical lysosomal enzyme that kills and digests microbial pathogens [15]. AKP is a hydrolase enzyme responsible for removing phosphate groups from many types of molecules, and capable of promoting phagocytosis through modification of the pathogen surface molecules [15,29]. LZM plays an important role in the innate immunity of fish, by lysis of bacterial cell wall peptidoglycans and stimulating the phagocytosis of bacteria [9]. C3 is most abundant in the complement system, assisting antibodies in recognizing pathogens, promoting phagocytosis, and lysis of target cells [15]. IgM is an important immunoglobulin and the only immunoglobulin class responding to antigenic challenges in both systemic and mucosal compartments in fish [15,30]. In the present trial, dietary CAs supplementation increased the activities of LZM, ACP, and AKP, and levels of C3 and IgM, in juvenile American eel, which suggests that the immune function may be enhanced by CAs supplementation. It was found that CAs may participate in immune regulation by promoting the development of immune organs, improving the activity of immune substances, and enhancing the expression of anti-inflammatory factors [31–33].

In this study, dietary CAs supplementation did not significantly affect the moisture, crude protein, crude lipid, or crude ash in the whole fish. Similar to the results of this trial, the body composition of whole fish was not significantly affected by 4.0 g/kg CAs (CAs A: formic acid, ammonium formate, and propionic acid or CAs B: benzoic acid, fumaric acid, and hydroxy analog of methionine) in olive flounder [10] and 2.0–4.0 g/kg CAs (fumaric acid, benzoic acid, and 2-hydroxy-4-methylthio butanoate acid) in yellow catfish [13]. In contrast to the present study, dietary supplementation with 5.0–15.0 g/kg CAs (calcium lactate and sodium acetate) increased crude protein and decreased crude lipid of whole Nile tilapia [8], whereas dietary supplementation with 2.0 g/kg CAs (formic acid, propionic acid, and calcium propionate) improved the levels of crude protein and crude lipid of whole Nile tilapia [7].

In this trial, appropriate dietary CAs supplementation significantly increased the contents of Ca and P in whole fish and accumulation rates of dietary Ca and P, which indicated that CAs could promote the absorption of dietary Ca and P in juvenile American eel. Similar to the results of the present trial, some previous studies revealed that 8.0–30.0 g/kg citric acid supplementation in a high soybean meal diet improved the P content of whole large yellow croaker (*Larimichthys crocea*) [34], and dietary 10 g/kg lactic acid supplementation in low phosphorus diets increased the accumulation and absorption of phosphorus in red sea bream [35]. This may be due to the ability of the CAs to dissociate into H+ and organic acid ions in the intestine of fish, where the lower pH may help in dissolving insoluble tricalcium phosphate in the diet, and certain organic acid anions are also able to combine with Ca and P to form complexes that are easily absorbed by the intestine [3,6,36]. However, dietary supplementation with 2.0–4.0 g/kg CAs (fumaric acid, benzoic acid, and 2-hydroxy-4-methylthio butyric acid) in the diet of yellow catfish did not significantly affect the Ca and P content of the whole fish [13].

It is worth noting that dietary supplementation with 5 g/kg CAs of juvenile American eel had certain negative effects on the growth, serum biochemical parameters, and accumulation of Ca and P. In agreement with the present results, the negative effects of excess CAs supplementation (citric acid, lactic acid, and phosphoric acid) were also found on the growth and intestinal digestive enzyme activity of Japanese sea bass [4]. Similarly, sodium butyrate supplemented at excess levels was reported to be detrimental to the growth of Nile tilapia [37], and the intestinal digestive enzyme activity and intestinal immunity of grass carp (*Ctenopharyngodon idella*) [38]. In this regard, some studies attributed this effect to the severe decrease in the pH of the digestive tract, which may damage the gastrointestinal mucosa [3,6]. In addition, excess levels of CAs can reduce palatability and decrease feed intake [6].

In general, both the present trial and many previous studies confirmed that an appropriate level of CAs supplementation had positive effects on the growth, serum biochemical parameters, and utilization of dietary Ca and P. Based on the quadratic regression relationship between the CAs levels and WGR or FCR, the optimal level of CAs supplementation

was 3.5–3.7 g/kg for the juvenile American eel. On the contrary, some studies exhibited no significant effects of CAs on growth performance, serum biochemical parameters, or body composition. The inconsistent results might be associated with the differences in some factors, including the acidifiers, trial fish, and diets, such as the chemical nature of the CAs (type of acids and site of the main action, pK value, molecular weight, and minimum inhibitory concentration of acid), levels of CAs supplementation, proportions of different types of acid, diversity of fish species, dietary composition, and buffering capacity of feed [1,3,6].

## 5. Conclusions

In summary, an appropriate supplementation level of CAs in the juvenile American eel diet may promote the growth performance, decrease the levels of serum lipid, modify the parameters of the liver function and immune function in the serum, and improve the utilization of dietary Ca and P. Dietary CAs supplementation had no significant effects on the body composition of whole fish. Excessive supplementation of CAs may cause negative effects on growth performance, serum biochemical parameters, and utilization of calcium and phosphorus. The recommended level of CAs in the diet was 3.5–3.7 g/kg to achieve the optimal WGR and FCR for the juvenile American eel. The effects of CAs on intestinal health should be studied to clarify the mechanism of growth-promotion effects in the future.

**Author Contributions:** Data curation, M.Z. and S.Z.; Methodology, S.Z.; Supervision, S.Z.; Writing—original draft, M.Z., X.W. and S.Z.; Writing—review and editing, M.Z., X.W. and S.Z. All authors have read and agreed to the published version of the manuscript.

**Funding:** This research was supported by the earmarked fund for China Agriculture Research System of MOF and MARA (CARS-46), Regional Development Project for Science and Technology Plan Program of Fujian Province (2022N3002), and Open Fund of Engineering Research Center of the Modern Industry Technology for Eel, Ministry of Education of China (RE202111; RE202112).

**Institutional Review Board Statement:** The study was conducted according to the guidelines of the Declaration of Helsinki and was approved by the Animal Care Advisory Committee of Jimei University (Approval No. 2020-0916-001).

**Data Availability Statement:** The data used during the current study are available from the corresponding author on reasonable request.

**Acknowledgments:** The authors thank Minghao Wang for providing the compound acidifiers manufactured by Trouw Nutrition R&D, NUTRECO NEDERLAND B.V.

**Conflicts of Interest:** The authors declare no conflict of interest.

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
