# Peer review of "Effect of Dietary Compound Acidifiers Supplementation on Growth Performance, Serum Biochemical Parameters, and Body Composition of Juvenile American Eel (Anguilla rostrata)"

_fishes, doi:10.3390/fishes7040203_

Round 1
Reviewer 1 Report
The manuscript entitled “Effect of Dietary Compound Acidifiers Supplementation on Growth Performance, Serum Biochemical Parameters, and Body Composition of Juvenile American Eel (Anguilla rostrata)”, authored by Mingliang Zhang, Xinyi Wu, and Shaowei Zhai, deals with the evaluation of the supplementation effects of acidifier compounds on the growth performance, serum biochemical parameters, and body composition of the juvenile American eel. The manuscript contains truly interesting data that could make a serious contribution to the current state of the art. The manuscript presents the data in a very clear and intuitive manner, and the authors egregiously discuss their results, even comparing them with data previously reported in the literature.
However, some changes need to be made to the manuscript before it can be considered suitable for publication.
The authors provide an excessively long abstract in the manuscript. As clearly stated in the Journal guidelines, this section should be a maximum of 200 words. In addition, the authors erroneously elaborate on the methodologies used in this section, poorly describing the current state of the art, the main issues, and the focus of their article.
The keyword section would also need to be rewritten. A maximum number of 10 words can be entered in this section. Word choice should fall on terms not in the title, at most used in the abstract. The usefulness of keywords is to facilitate the search of the manuscript after publication using the most common scientific search motives. Consequently, I strongly suggest that authors eliminate words already used in the title, and replace them with other words that can make their article stand out after publication.
The introduction and discussion section are really well written. The authors propitiously make comparisons with data previously published in the literature, and best describe the data they obtained. A peakal request I would make of the authors, since the Journal does not charge extra for publishing colored images, I would suggest that the authors transform their graphics. The use of color helps comprehension, and promotes reader understanding (optional).
Equation reported in 2.4. section should me placed in a table.
The conclusion section should be rewritten, adding a small description of the results obtained and possible future prospects.
Author Response
The manuscript entitled “Effect of Dietary Compound Acidifiers Supplementation on Growth Performance, Serum Biochemical Parameters, and Body Composition of Juvenile American Eel (Anguilla rostrata)”, authored by Mingliang Zhang, Xinyi Wu, and Shaowei Zhai, deals with the evaluation of the supplementation effects of acidifier compounds on the growth performance, serum biochemical parameters, and body composition of the juvenile American eel. The manuscript contains truly interesting data that could make a serious contribution to the current state of the art. The manuscript presents the data in a very clear and intuitive manner, and the authors egregiously discuss their results, even comparing them with data previously reported in the literature.
However, some changes need to be made to the manuscript before it can be considered suitable for publication.
The authors provide an excessively long abstract in the manuscript. As clearly stated in the Journal guidelines, this section should be a maximum of 200 words. In addition, the authors erroneously elaborate on the methodologies used in this section, poorly describing the current state of the art, the main issues, and the focus of their article.
Re: Many thanks for your advices, the word number of our abstract is less than 200 words now, and we focused the main issues of compound acids and results of the feeding trial. Please see the revised manuscript.
The keyword section would also need to be rewritten. A maximum number of 10 words can be entered in this section. Word choice should fall on terms not in the title, at most used in the abstract. The usefulness of keywords is to facilitate the search of the manuscript after publication using the most common scientific search motives. Consequently, I strongly suggest that authors eliminate words already used in the title, and replace them with other words that can make their article stand out after publication.
Re: Many thanks for your advices about the keyword, and the keyword section was rewritten. We added several keyword in the revised manuscript according to your suggestions.
The introduction and discussion section are really well written. The authors propitiously make comparisons with data previously published in the literature, and best describe the data they obtained. A peakal request I would make of the authors, since the Journal does not charge extra for publishing colored images, I would suggest that the authors transform their graphics. The use of color helps comprehension, and promotes reader understanding (optional).
Re: Thank you for the advices, the present status of the figures could show the results of this trial, we have to submit the revised manuscript, and the color images might take some time. So we did not transform the graphics.
Equation reported in 2.4. section should me placed in a table.
Re:Many thanks for your advices, we placed the equation in the note of table or figure. Please see the revised manuscript.
The conclusion section should be rewritten, adding a small description of the results obtained and possible future prospects.
Re:Thanks, we added some description of the results and possible future prospects in the revised manuscript.
Reviewer 2 Report
The manuscript is interesting and well written. I only found a few minor issues listed herein
Line 63 “did” instead of “could”
Line 69 “is” instead of “was”
Line 71 "ginseng in water" please explain the meaning
Line 231 macroelements instead of microelements
Line 261 “are” instead of “were”
Lines 268-273 the phrase lacks the verb. Please reformulate
Lines 259 and 328 What do you mean with “generally”?
Line 374 “on the contrary” instead of “while”
Author Response
Re: many thanks for your advices to improve the quality of our manuscript. we corrected all the problem you listed, please see the revised manuscript.
Line 63: The “could” was replaced by “did” in the revised manuscript.
Line 69:The “is” was replaced by “was” in the revised manuscript.
Line 71: "ginseng in water" is used as a metaphor to imply that there is very important medicine effect if you often eat eel, it is just like eating the ginseng to make you more healthy.
Line 231 : The “macroelements” was replaced by “microelements” in the revised manuscript.
Line 261 : The “are” was replaced by “were” in the revised manuscript.
Lines 268-273: we rewrote the sentence according to your suggestion.
Lines 259 and 328: the “generally” was deleted in the revised manuscript, which did not affect the meaning of those sentences.
Line 374 : The “while” was replaced by “on the contrary” in the revised manuscript.
Reviewer 3 Report
This manuscript provides a very interesting area of work that has an unusual species for target. The choice of eel is most timely and relevant for China as it is lucrative and of high value. The quest for improved diets for aquaculture and farmed fish is in demand to make for healthier animals and to reduce disease through prophylactic dietary additives and supplements. This has been investigated to good effect by the authors through the choice of acidifiers as the topic. The work plan is very comprehensive, and well executed in terms of the science an approach. It is a classical type of study where a series of diets based on an appropriate baseline formulation meeting the nutritional requirements of the eel as far as we know. The work embraces several aspects beyond growth and feed utilization. There is data for serum parameters relating to health status and linked to the overall nutritional and health performance of the eel. It would however have been very interesting to have reported the condition of the gut and level of integrity as relating to the effects of the acidifiers used. some inclusion of gut histology would have been very nice however you have undertaken much work to present much data concerning the application of acidifiers on a complete spectrum of relevant enzymes and associated metabolites. Your introduction is very well argued and logical in its examination of the literature base. The rationale of the investigation is well explained and the scientific formatting of the plan of work good. The results were clear and the statistical applications appropriate. The tables, figures and legends do justice and these were sequentially presented. The discussion was able to convey the basis of the finding and critically integrate with the scientific literature to provide a good overview. In summary I do like this manuscript. It gives new and unusual insights on the use of acidifiers to eels that are important in China. Only aspect is to avoid over emphasis of some expressions such as in abstract ..( e.g. there are more and more reports) keep it more professionally stated)
Author Response
Re: many thanks for your conformation and appreciation about our work. The aim of present manuscript was to present the beneficial effects of appropriate level of compound acidifiers on growth, serum biochemical parameters, and accumulations of Ca and P. The results of intestinal health including the digestive enzymes, the integrity, histology, microbiota community, metabolites of intestine are currently being measured and prepared. We wanted to publish those results in another paper.
We rewrote the abstract according to your advices, please the revised manuscript.